# Under-five mortality and associated factors in southeastern Ethiopia

**Firaol Lemessa Kitila**[1,2], **Rahel Milkias Petros**[3], **Gebi Hussein Jima**[2], **Tewodros Desalegn**[2], **Abebe Sorsa**[3], **Isaac Yaw Massey**[1], **Chengcheng Zhang**[1], **Fei Yang**[1]*

1 Department of Occupational and Environmental Health, Xiangya School of Public Health, Central South University, Changsha, Hunan Province, China, 2 Department of Public Health, College of Health Sciences, Arsi University, Asella, Oromia, Ethiopia, 3 Department of Pediatrics and Child Health, Asella Teaching and Referral Hospital, College of Health Sciences, Arsi University, Asella, Oromia, Ethiopia

* phfyang@csu.edu.cn

## Abstract

### Background

In the year 2019, around 5 million children under age five died and most of the deaths happened in developing countries. Though large numbers of deaths are reported in such countries, limited availability of data poses a substantial challenge on generating reliable estimates. Hence, this study aims to assess the prevalence and factors associated with under-five mortality in southeastern Ethiopia.

### Methods

A register based cross sectional study was conducted from 1st September 2014 to July 2019 in Asella teaching and referral hospital. A total of 4901 under-five age children registered on the admission and discharge book of pediatric ward with complete information were included for the analysis. Data entry and analysis were conducted using Epidata Version 7 and SPSS version 21, respectively. Descriptive statistics were used to explore the characteristics of the study participants and their condition at discharge. Adjusted Odds Ratio (AOR) with its 95% Confidence interval and P-value less than 5% was used to decide the statistically significant association.

### Results

The prevalence of under-five mortality among admitted children in Asella Teaching and Referral hospital was 8.7% (95% CI 7.91–9.50%). Post-Neonatal and Child mortality were found to be 9.1% and 8.18%, respectively. Moreover, large numbers of death (45.2%) were seen within the first 2 days of admission. Address (AOR:1.4(1.08–1.81)), HIV status (AOR:4.64 (2.19–9.8)), severe acute malnutrition (AOR:2.82 (2.03–3.91)), hypovolemic shock (AOR:4.32 (2.31–8.1)), type I diabetes with DKA (AOR:3.53(1.34–9.29) and length of stay in the hospital for ≤2 days (AOR: 4.28 (3.09–5.95)) as well as 3–4 days (AOR: 1.48 (1.02–2.15)) were among the identified predictors.

**Data Availability Statement:** All relevant data are within the manuscript and its Supporting Information files.

**Funding:** The authors received no specific funding for this work.

**Competing interests:** The authors have declared that no competing interests exist.

## Conclusions

Though childhood mortality is swiftly decreasing, and access and utilization of health care is improving in Ethiopia, our study found large prevalence of under-five mortality, 8.7% and higher number of deaths in early days of admission. Improving the quality of service has a paramount importance in reducing the mortality and managing associated factors contributing to under-five mortality among admitted children.

## Introduction

The third Sustainable and Development Goals (SDG) was set by United Nation (UN) to safeguard healthy lives and promote wellbeing for all individuals irrespective of their age. Besides, it calls for an end to avoidable deaths of children, with all countries aiming to decrease under-five mortality; the risk of child dying before completing five years of age, to at least as low as 25 deaths per 1,000 live births by 2030 [1, 2].

Though the global number of child deaths remains high, the world has made remarkable progresses in reducing child and young adolescent mortality over the past few decades. The global under-five mortality rate dropped by 59% from 93 deaths per 1,000 live births in 1990 to 39 in 2018, while mortality among children aged 5–14 years fell by 53% from 15 to 7 deaths per 1000 children [3]. In spite of the progress over the past decades, in 2019 alone, around 5 million children under age 5 died [4].

Ethiopia has also achieved significant decline in under-five mortality over recent decades. Yohannes et al. reported that under-five mortality in Ethiopia was 72% lower in 2015 than it was 25 years ago [5]. Despite significant progress in reducing child deaths, children from rural domiciliary remain significantly vulnerable, as child survival interventions are not reaching the children who need them most. Research indicates that, delays in seeking assessment and treatment as well as malnutrition are among the factors that exacerbate child mortality [6–8]. Furthermore, early child death (death within ≤24 hours of arrival in the hospital) is commonly caused by preventable and reversible diseases such as pneumonia, sepsis, malaria, heart failure, anemia, acute respiratory tract infection and diarrhea [9].

Despite the achievement, the limited availability of data in southeastern Ethiopia poses a considerable challenge on generating reliable estimates of under-five mortality in that particular area. Identifying the prevalence and specific determinants of under-five mortality are a key in planning and implementing interventions [10]. Therefore, this study aims to identify the overall prevalence and associated risk factors for under-five, child and post-neonatal mortality in Asella teaching and referral hospital, southeastern Ethiopia.

## Materials and methods

### Study design, setting, and participants

A register based cross sectional study was conducted from 1st September 2014 to July 2019 in Asella teaching and referral hospital, located about 175KM from the capital, Addis Ababa. The hospital is the only teaching and referral center in the region of southeastern Ethiopia and the total population served in the catchment area is about 3,459,322 [11].

A total of 4901 under-five age children registered on the admission and discharge book of pediatric ward with complete information (data filled with all information on the registration

book) were recruited to the study. Cases admitted in wards other than pediatrics and with incomplete information were excluded from the study.

### Data collection tool and quality control methods

Structured data sheet was prepared and used to extract data of the eligible cases from the registration book of pediatrics ward. To ensure quality of the data, training was given to two nurses before they began the extraction, and the data collection process was conducted under the close supervision of the study team. In addition, the collected data were verified for completeness and the supervision team also took 10% of the collected data randomly and cross-checked it with the registration book.

### Data processing and analysis

The collected data on the sheet from the hand written hospital registry was entered into Epidata version 7 and analysis were carried out using SPSS version 21. Descriptive statistics were used to explore the characteristics of the study participants and their condition at discharge. Bivariate logistic regression was conducted to select candidate variables for multivariate analysis: variables found to have association with the dependent variable at 0.2 P-value were entered in to multivariate logistic regression for controlling the possible effect of confounders. Adjusted Odds Ratio (AOR) with its 95% confidence interval and P-value less than 5% was used to decide the statistically significant association.

### Variables

Dependent (outcome) variable: Under-five mortality (condition at discharge).

Independent variables (all were taken from registration book):

Socio-demographic characteristics like child's age, Sex, Place of residence

Child related issues such as National classification of disease diagnosed (Acute Gastro-Enteritis with Dehydration, Anemia, Bronchiolitis, Congestive heart disease, Congenital Anomaly, CROUP, epilepsy, Foreign body, Hyper-Active Airway Disorder (HAAD), Measles, Meningitis, Pertussis, Severe Acute Malnutrition (SAM), Sepsis, Severe Pneumonia, Hypovolemic Shock, Surgical Conditions, Tuberculosis, Type I diabetes with DKA, Urinary Tract Infections and Other conditions), HIV test result, TB screening result, Date of admission (DD/MM/YYYY), Date of discharge (DD/MM/YYYY) and length of stay (admission) in days.

### Operational definitions

Under-5 Mortality: The probability of dying before the fifth birthday.

Condition at Discharge: Discharge conditions on the registration book which were classified as Alive (Improved, Same, Left Against Medical (Self-discharge), Referred to higher (Referred for diagnostic or therapeutic services that are not available at the hospital OR referred because the hospital beds are all occupied) and Absconded (Escaped)) and Died.

National classification of diseases: Primarily diagnosed diseases at the time of admission in the hospital. It contains all diagnosed cases like SAM, Measles, meningitis, acute gastro-enteritis with dehydration, surgical conditions and others.

Urban residents: Individuals living in cities (which have a population of at least 50,000 inhabitants in contiguous dense grid cells with >1,500 inhabitants per $km^2$) and towns which have a population of at least 5,000 inhabitants in contiguous dense grid cells with >300 inhabitants per $km^2$).

Rural residents: Individuals residing mostly of low-density grid cells.

### Ethical approval and consent to participate

Ethical approval was obtained from Ethical Review Board of college of health sciences, Arsi University (Project Protocol Number—A/CHS/RC/14/2019). Permission was sought from asella teaching and referral hospital to obtain and use the secondary data from registration book for this study. As per Arsi University college of health science ethical review board, consent of patient/guardian is not needed for secondary data research as it is not feasible to access them. Hence, the requirement for obtaining consent was waived and all data were collected in accordance with the relevant guidelines and regulations (Declaration of Helsinki). However, confidentiality of information retrieved from patients' medical records was upheld by excluding the participant identifiers (name and patient record number).

## Results

### 1. Characteristics of the study participants

Table 1 shows basic characteristics of the study participants. Over the years from 1st September 2014 to July 2019, 4901 admitted under-five children in pediatric ward of Asella referral and teaching hospital with complete information were included in the study (14 under-five children with incomplete data were excluded). The minimum and maximum age of the study participants were, 1 and 59 months, respectively. The average length of stay being admitted at the hospital was 6.72 days with 127 days as maximum, and 1 day as minimum stay. Among the admitted children, males constituted 60.6%, 58.4% were in the age range of 1–12 months and 74.9% were rural residents. Moreover 0.8% of the admitted children were positive for HIV and the condition of 86.1% improved at discharge.

### 2. National classification of disease

In Table 2, national classification of disease with discharge condition of the children is shown. Among the primary diagnosis at the time of admission, severe pneumonia accounted for 34.2%, followed by malnutrition and acute gastro-enteritis with dehydration, 15.8% and 10.7%, respectively. Cases included under surgical conditions accounted for over 10% and more than 17 diseases were categorized under other conditions among which poliomyelitis accounted for 2 cases (both cases were diagnosed in 2017) during the five years' study period. In addition, 290 cases were diagnosed with meningitis, 40 with tuberculosis, 57 with different degrees of anemia and 35 with congenital abnormality. During the study period, vaccine preventable diseases like measles and whooping cough accounted for 0.75% and 0.98% of cases, respectively.

### 3. Prevalence of under-five, post-neonatal and child mortality

Under-Five, Post-Neonatal and Child mortality is summarized in Table 3. Among 4901 admitted children with different clinical diagnosis from September 2014 to July 2019, 427 (8.7% (95% CI 7.91%-9.50%)) died. Post-Neonatal and Child mortality among admitted children were found to be 91/1000 and 81.8/1000, respectively.

### 4. Outcome of admitted children along with length of admission

Fig 1 shows discharge condition along with length of admission at hospital. Almost half of the admitted children (48%) stayed in the hospital for 4 or less days from which 45.2% (193) died in early days of admission (1–2 days). Among those who died, 17.6% of death happens in between 3–4 days of admission.

**Table 1. Characteristics of the study participants admitted at Asella Referral & Teaching Hospital, January, 2020 G.C.**

| Characteristics | Number | Percent |
|---|---|---|
| Sex | | |
| Male | 2968 | 60.6 |
| Female | 1933 | 39.4 |
| Age in months | | |
| 1–12 | 2859 | 58.4 |
| 13–24 | 990 | 20.2 |
| 25–36 | 452 | 9.2 |
| 37–48 | 363 | 7.4 |
| 49–59 | 237 | 4.8 |
| Place of residence | | |
| Urban | 1229 | 25.1 |
| Rural | 3672 | 74.9 |
| HIV Screening result | | |
| Positive | 40 | 0.8 |
| Negative | 4861 | 99.2 |
| Year of admission in G.C. | | |
| Sep, 2014-Aug, 2015 | 1079 | 22.0 |
| Sep, 2015-Aug, 2016 | 1098 | 22.4 |
| Sep, 2016-Aug, 2017 | 941 | 19.2 |
| Sep, 2017-Aug, 2018 | 1073 | 21.9 |
| Sep, 2018-Jul, 2019 | 710 | 14.5 |
| Length of stay in hospital | | |
| ≤2 days | 1176 | 24.0 |
| 3–4 days | 1180 | 24.1 |
| 5–7 days | 1016 | 20.7 |
| 8–10 days | 636 | 13.0 |
| >10 days | 893 | 18.2 |
| Condition at Discharge | | |
| Improved | 4219 | 86.1 |
| Same | 29 | 0.6 |
| Self-Discharge | 91 | 1.9 |
| Died | 427 | 8.7 |
| Referred | 131 | 2.7 |
| Absconded | 4 | 0.1 |

## 5. Bivariate and multivariate analysis

Table 4 presents the crude and adjusted odds ratio for various factors associated with condition at discharge among admitted children. Overall, place of residence, HIV status, severe acute malnutrition, hypovolemic shock, type I diabetes with DKA and length of stay in the hospital for ≤2 days and 3–4 days compared with those who stayed more than 10 days were among factors that were significantly associated with under-five mortality. Besides, severe pneumonia was also identified as a predictor of under-five mortality.

## Discussion

Given that infant and child mortality rates are the basic indicators of a country's socioeconomic situation and quality of life [12], this study was conducted to assess the prevalence and

**Table 2. National classification of disease with discharge condition for under five children admitted at Asella Referral and Teaching Hospital, January 2020.**

| National classification of diseases (Primary Diagnosis at Admission) | Condition of Discharge | | Total |
|---|---|---|---|
| | Alive | Died | |
| | Freq (%) | Freq (%) | Freq (%) |
| Acute Gastro-Enteritis with Dehydration | 488 (9.96) | 38 (0.77) | 526 (10.73) |
| Anemia | 53 (1.08) | 4 (0.08) | 57 (1.16) |
| Bronchiolitis | 39 (0.8) | 3 (0.06) | 42 (0.86) |
| Congestive heart disease | 96 (1.96) | 12 (0.24) | 108 (2.2) |
| Congenital Anomaly | 31 (0.63) | 4 (0.08) | 35 (0.71) |
| CROUP | 54 (1.1) | 2 (0.04) | 56 (1.14) |
| Epilepsy | 43 (0.88) | 3 (0.06) | 46 (0.94) |
| Foreign Body | 44 (0.9) | 2 (0.04) | 46 (0.94) |
| Hyper-Active Airway Disorder (HAAD) | 72 (1.47) | 6 (0.12) | 78 (1.59) |
| Measles | 34 (0.69) | 3 (0.06) | 37 (0.75) |
| Meningitis | 259 (5.28) | 31 (0.63) | 290 (5.92) |
| Pertussis | 46 (0.94) | 2 (0.04) | 48 (0.98) |
| Severe acute malnutrition (SAM) | 654 (13.34) | 121 (2.47) | 775 (15.81) |
| Sepsis | 188 (3.83) | 20 (0.41) | 208 (4.24) |
| Severe Pneumonia | 1573 (32.1) | 105 (2.14) | 1678 (34.24) |
| Hypovolemic Shock | 41 (0.84) | 18 (0.37) | 59 (1.2) |
| Surgical conditions | 472 (9.63) | 24 (0.49) | 496 (10.12) |
| Tuberculosis | 35 (0.71) | 5 (0.1) | 40 (0.82) |
| Type I Diabetes with DKA | 23 (0.5) | 6 (0.12) | 29 (0.59) |
| Urinary Tract Infection | 51 (1.04) | 1 (0.02) | 52 (1.06) |
| Other conditions | 178 (3.63) | 17 (0.35) | 195 (3.98) |

(Other medical conditions included are Alcohol Intoxication, Animal Bite, Arthritis, Chicken Pox, Hypertension, Hydrocele, Hypoglycemia, Mumps, Osteomyelitis, Otitis Media, Poisoning, Poliomyelitis, Rabies, Rectal Prolapse, Rickets, RVI, Sexual Assault and Acute Viral Hepatitis AND cases included under Surgical conditions are abscess, acute abdomen, burn, cellulitis, trauma, hernia, hydrocele, animal bite and rectal prolapse).

associated factors for under-five, child and post-neonatal mortality in Asella teaching and referral hospital, southeastern Ethiopia. The study revealed the prevalence of 8.7% (87/1000) (95% CI, 7.91%-9.50%) under-five mortality. Post-neonatal and child mortality rates were 9.1% (91/1000) and 8.18% (81.8/1000), respectively. Place of residence, HIV status, severe acute malnutrition, hypovolemic shock, type I diabetes with DKA and length of stay in the hospital for ≤2 days and 3–4 days were among the identified predictors.

**Table 3. Under-five, post-neonatal and child mortality among children admitted to pediatrics ward of Asella Teaching and Referral Hospital, January 2020 G.C.**

| Age of the child | Discharge condition | Freq | % |
|---|---|---|---|
| Under-Five Children | Alive | 4474 | 91.3 |
| | Died | 427 | **8.7** |
| 1–12 Months (Post-Neonatal) | Alive | 2599 | 90.9 |
| | Died | 260 | **9.1** |
| 13–59 Months (Child) | Alive | 1875 | 91.82 |
| | Died | 167 | **8.18** |

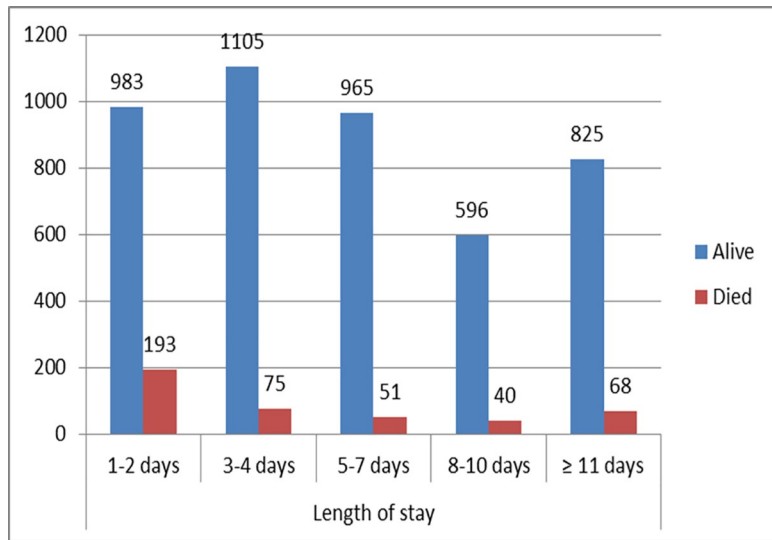

**Fig 1. Discharge condition of children along with the length of stay in Asella Teaching & Referral Hospital, 2020.**

The finding of this study was higher compared with Ethiopian Demographic and Health Survey (EDHS) 2019 report. As per the report, the overall under-five mortality, child mortality and post-neonatal mortality rate was 55, 12 and 13 deaths per 1,000 live births, respectively [2]. This might be due to population used as a denominator and the study setting. Using children diagnosed and admitted to the hospital as a denominator could be the reason for higher level of occurrence of post-neonatal, child and under-five mortality in our study. On the other hand, different studies conducted on under-five mortality demonstrated lower mortality rates compared with our study. Observations from the studies conducted in Ghana and Namibia showed under-five mortality of 3.62% and 1.2%, respectively [13, 14]. Besides, cohort study on predictors of post neonatal mortality in Western Kenya indicated 4.6% post-neonatal mortality [15]. It is on interest that the study design and setting may have accounted for these differences. Even though the ultimate outcome of sick children depends on different factors, higher deaths in teaching and referral hospitals might be attributed to more critical cases referred from district hospitals and health centers at critical hours. Contrary to our finding, study conducted on epidemiology of child mortality in a tertiary hospital in India revealed 10% of death among admitted children [16]. Moreover, there is still high burden of under-five mortality in many countries in West African region [17]. The difference could also be attributed to the study design and setting.

The magnitude and severity of child mortality are exacerbated by different factors. In our study, children residing in rural areas were at higher risk of mortality compared with those residing in urban areas. Survival analysis of under-five mortality in Nigeria also depicted a consistent result in which place of residence significantly contributed to the risk of under-five mortality [6]. There are more evidences indicating residing in rural area as a predictor of under-five mortality in different studies [7, 18]. This could be related with poor child care practices, unavailability of suitable medical access, poor transportation system, lower awareness of community on health and delays in health care seeking behavior of families residing in rural area.

Even though starting ART instantly to all children living with HIV regardless of their WHO clinical stages and CD4 counts/percentage is recommended and being implemented [19], our study depicted HIV as a significant predictor for under-five mortality. This finding was

**Table 4. Crude and Adjusted odds ratios for various factors in relation with condition at discharge among under-five children in Asella Teaching and Referral Hospital, January 2020.**

| Variables | Condition at Discharge | | Crude OR (95% CI) | Adjusted OR (95% CI) |
|---|---|---|---|---|
| | Alive | Died | | |
| **Place of residence** | | | | |
| Urban | 1142 | 87 | 1 | 1 |
| Rural | 3332 | 340 | 1.34 (1.05–1.71) | 1.39 (1.08–1.8) * |
| **HIV Status** | | | | |
| Negative | 4445 | 416 | 1 | 1 |
| Positive | 29 | 11 | 4.05 (2.01–8.17) | 4.64 (2.19–9.8) *** |
| **Length of stay** | | | | |
| ≤2 days | 983 | 193 | 2.38 (1.78–3.19) | 4.28 (3.09–5.95) *** |
| 3–4 days | 1105 | 75 | 0.82 (0.59–1.16) | 1.48 (1.02–2.15) * |
| 5–7 days | 965 | 51 | 0.64 (0.44–0.93) | 0.99 (0.67–1.46) |
| 8–10 days | 596 | 40 | 0.81 (0.54–1.22) | 0.90 (0.60–1.35) |
| >10 days | 825 | 68 | 1 | 1 |
| **Acute Gastro-Enteritis with Dehydration** | | | | |
| No | 3986 | 389 | 1 | 1 |
| Yes | 488 | 38 | 0.8 (0.56–1.13) | 0.68 (0.45–1.03) |
| **CROUP** | | | | |
| No | 4420 | 425 | 1 | 1 |
| Yes | 54 | 2 | 0.39 (0.09–1.59) | 0.24 (0.06–1.04) |
| **Meningitis** | | | | |
| No | 4215 | 396 | 1 | 1 |
| Yes | 259 | 31 | 1.27 (0.87–1.87) | 1.45 (0.92–2.27) |
| **Severe Acute Mal-nutrition** | | | | |
| No | 3820 | 306 | 1 | 1 |
| Yes | 654 | 121 | 2.31 (1.84–2.90) | 2.82 (2.03–3.91) *** |
| **Severe Pneumonia** | | | | |
| No | 2901 | 322 | 1 | 1 |
| Yes | 1573 | 105 | 0.60 (0.48–0.76) | 0.71 (0.52–0.97) * |
| **Hypovolemic Shock** | | | | |
| No | 4433 | 409 | 1 | 1 |
| Yes | 41 | 18 | 4.76 (2.71–8.36) | 4.32 (2.31–8.1) *** |
| **Surgical condition** | | | | |
| No | 4002 | 403 | 1 | 1 |
| Yes | 472 | 24 | 0.51 (0.33–0.77) | 0.64 (0.39–1.03) |
| **Type I DM with DKA** | | | | |
| No | 4451 | 421 | 1 | 1 |
| Yes | 23 | 6 | 2.76 (1.12–6.81) | 3.53 (1.34–9.29) * |
| **Urinary Tract Infection** | | | | |
| No | 4423 | 426 | 1 | 1 |
| Yes | 51 | 1 | 0.20 (0.03–1.48) | 0.25 (0.03–1.88) |

* is $p < 0.05$

** is $p < 0.01$

*** is $p < 0.001$.

consistent with a result from cohort study in Kenya and South Africa in which HIV infection was associated with increased mortality [15, 20]. The evidence suggests that HIV-infection may increase susceptibility to different diseases and greatly impact the health of HIV-infected

children, which subsequently can increase the occurrence of under-five mortality. Among the participants, our study found HIV prevalence of 0.8%, nearly close to the nationwide HIV prevalence (0.9%). The data obtained from UNAIDS in 2019 also showed that approximately 44,000 children less than 14 years lived with HIV [21].

In terms of length of stay in hospital, our study indicated higher number of deaths in ≤2 days of stay, 45.2% and 3–4 days of stay, 17.6% in comparison with those admitted and stayed for more than 10 days. A consistent finding was seen in a study on epidemiology of child mortality in India in which higher number of deaths, 68% occurred in the first 48 hours [16]. Similarly, a finding from the study in Namibia revealed death of 51.7% of children within the first three days of admission [14]. This doesn't mean length of admission in the hospital is a predictor of mortality but rather, delays in care seeking by family members and late referral of the sick children might contribute for higher death in early days of admission. In addition, the priority and quality of emergency care given to the cases in the hospital might also be a reason. The present study also identified some proximate diseases and other health condition as predictors of under-five mortality. Shock and severe acute malnutrition were among the identified predictors. This finding was consistent with the result from a study conducted in Ghana and Northwestern Ethiopia, which identified malnutrition and shock as a predictor, respectively [8, 22]. The high mortality occurring with severe acute malnutrition or shock could be explained by the poor reaction of the body that has suffered reductive adaptation to the medical intervention at the time of admission in the hospital.

Among the study participants, pneumonia accounted for 1678 (34.24%) from which 105 died. The magnitude of pneumonia in our finding was consistent with the result from a systematic review in east Africa (34%) and study in Wondo Genet district (33.5%) [23, 24]. In contrast, lower prevalence of pneumonia was reported in a review conducted in Ethiopia (20.68%) and study conducted in Arsi zone (17.7%) [25, 26]. Although pneumonia accounted for 24.6% (105) of total deaths in the hospital, it was found among the factors that have protective effect from the death in under-five children (OR: 0.69 CI: 0.52–0.91). Unlike our finding, the study on hospitalized children at a referral hospital in Addis Ababa identified pneumonia as a cause of mortality in all age groups [27]. Similarly, a study in Limpopo province of South Africa and Kassena-Nankana District of Northern Ghana depicted lower respiratory tract infection as a predictor for under-five mortality [20, 28]. The difference might be because the health care workers in the hospital had better awareness in managing severe pneumonia as the numbers of diagnosed cases were very large and repeated exposure could make them manage the cases better. Above and beyond, such cases are usually treated with priority and the health care workers give due emphasis.

The dreadful thing in our study area is having a report on vaccine preventable diseases such as pertussis, measles and polio. Similar finding was seen in a study from Sokoto state Nigeria [29]. For diseases that are so rare in a given population like polio, occurrence of a single case is an indication for large number of asymptomatic children in the community. Reason for the occurrence of these vaccine preventable diseases is mainly attributed to vaccination coverage; in our case, 18.9% of children residing in the region where our hospital belongs had no vaccination at all [2].

We acknowledge that our study has a number of caveats. To mention some, the finding should be considered cautiously while interpreting since the data were retrospectively extracted from hand written medical records. In addition, under-five children admitted outside the pediatric ward of the hospital were not included in the study and the cause for under-five mortality made by the health care providers may not accurately reflect the entire cause of death provided that autopsy is not performed. Moreover, our study did not account for any gaps in the quality of clinical care which may have contributed to high rates of mortality.

Despite these caveats, we used a large sample size and believe that this study will provide valuable information for researchers, the hospital administration, ministry of health and other concerned stake holders.

## Conclusion

Though childhood mortality is swiftly decreasing, and access as well as utilization of health care is improving in Ethiopia, our study found large prevalence of under-five mortality, 8.7% (87/1000). In addition, post-neonatal and child mortality were 9.1% and 8.18%, respectively. Place of residence, HIV status, length of stay, severe acute malnutrition, hypovolemic shock and type I diabetes with DKA were among the identified predictors. From total death, more than 45% occurred within the first 48 hours of stay in the hospital. This condition demands awareness creation on early health care seeking behavior, a further review of the referral system, implementation of proper triage, early screening for HIV and improving quality of care provided for sick children. Besides, we don't need to get complacent with significant decline in under-five mortality over recent decades in the country as we are still having higher mortality rate that demands attention.

## Supporting information

**S1 File.**
(SAV)

## Acknowledgments

We would like to express our sincere gratitude to Asella Teaching and Referral Hospital for granting access to the hospital records.

## Author Contributions

**Conceptualization:** Firaol Lemessa Kitila, Rahel Milkias Petros, Gebi Hussein Jima, Tewodros Desalegn, Abebe Sorsa, Isaac Yaw Massey, Chengcheng Zhang, Fei Yang.

**Data curation:** Firaol Lemessa Kitila, Rahel Milkias Petros, Gebi Hussein Jima, Fei Yang.

**Formal analysis:** Firaol Lemessa Kitila, Rahel Milkias Petros, Gebi Hussein Jima, Abebe Sorsa, Isaac Yaw Massey, Chengcheng Zhang, Fei Yang.

**Investigation:** Firaol Lemessa Kitila, Rahel Milkias Petros, Fei Yang.

**Methodology:** Firaol Lemessa Kitila, Rahel Milkias Petros, Gebi Hussein Jima, Tewodros Desalegn, Abebe Sorsa, Isaac Yaw Massey, Chengcheng Zhang, Fei Yang.

**Project administration:** Rahel Milkias Petros.

**Resources:** Firaol Lemessa Kitila, Rahel Milkias Petros.

**Software:** Firaol Lemessa Kitila, Rahel Milkias Petros, Tewodros Desalegn, Fei Yang.

**Supervision:** Firaol Lemessa Kitila, Rahel Milkias Petros, Fei Yang.

**Validation:** Firaol Lemessa Kitila, Rahel Milkias Petros.

**Writing – original draft:** Firaol Lemessa Kitila, Rahel Milkias Petros, Gebi Hussein Jima, Tewodros Desalegn, Abebe Sorsa, Isaac Yaw Massey, Chengcheng Zhang, Fei Yang.

**Writing – review & editing:** Firaol Lemessa Kitila, Rahel Milkias Petros, Gebi Hussein Jima, Tewodros Desalegn, Abebe Sorsa, Isaac Yaw Massey, Chengcheng Zhang, Fei Yang.

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
