## [Decision Letter · Decision Letter 0]

8 Apr 2021

PONE-D-21-06094

Under-Five Mortality and Associated Factors in Southeastern Ethiopia

PLOS ONE

Dear Dr. Yang,

Thank you for submitting your manuscript to PLOS ONE. After careful consideration, we feel that it has merit but does not fully meet PLOS ONE’s publication criteria as it currently stands. Therefore, we invite you to submit a revised version of the manuscript that addresses the points raised during the review process.

We look forward to receiving your revised manuscript.

Kind regards,

Tai-Heng Chen, M.D.

Academic Editor

PLOS ONE

4. Please include in your Methods section (or in Supplementary Information files) the participating hospitals/institutions.

Reviewers' comments:

Reviewer's Responses to Questions

**Comments to the Author**

1. Is the manuscript technically sound, and do the data support the conclusions?

Reviewer #1: Yes

Reviewer #2: Partly

Reviewer #3: Yes

2. Has the statistical analysis been performed appropriately and rigorously? 

Reviewer #1: Yes

Reviewer #2: No

Reviewer #3: Yes

3. Have the authors made all data underlying the findings in their manuscript fully available?

Reviewer #1: Yes

Reviewer #2: No

Reviewer #3: Yes

4. Is the manuscript presented in an intelligible fashion and written in standard English?

Reviewer #1: No

Reviewer #2: No

Reviewer #3: Yes

5. Review Comments to the Author

Reviewer #1: The authors report a cross-sectional study with nearly 5,000 pediatric hospital admissions in a single center in Ethiopia. They multiple clinical characteristics associated with mortality. The strengths of this study are the addressing of an important problem that is often not studied in this region, the use of basic and straightforward statistics, and the simple study design and approach to address a worthwhile question.

Despite the article’s strengths, there are several weaknesses that this reviewer thinks should be addressed. Some of these weaknesses include the lack of a clear description of many of the variables or the rationale for selecting these variables, the need to further contextualize the study’s findings, and the need to address missing data further. Another large limitation is the lack of laboratory data and lack of information on care received, or not received. Lastly, this article needs to be carefully reviewed by a native-English speaker. Specifically, the authors use the word “besides” incorrectly multiple times. I provided specific comments by section here below.

Introduction:

-The number of under-5 deaths globally needs to be either updated or cited correctly. The most recent Global Burden of Disease study from 2019 states this number is now closer to 5 million per year. (https://www.thelancet.com/journals/lancet/article/PIIS0140-6736(20)30977-6/fulltext)

-Please define what is meant by “early child mortality” used in paragraph 3.

-Paragraph 4: It’s not clear what the authors mean exactly by “the limited availability of data” as the prior 3 paragraphs provide estimates on under-5 mortality and even data on childhood mortality trends in Ethiopia.

Methods:

-I suggest the authors remove the mention of the period in which the study was conducted (i.e. August-November 2019) as it creates confusion. The study period appears to be Sept. 2014-July 2019 in paragraph 2.

-What do the authors mean by “complete information” in paragraph 2 of the Methods?

-The authors mention that patients without complete data were excluded. I suggest they mention what kind of bias may have been introduced in this in the Limitations.

-Did the authors electronically transcribe what was handwritten in the registry at this hospital? This needs to be clarified.

-I suggest the authors provide some rationale for the selection of their independent variables. Was there previous literature upon which the authors decided to select these candidate variables? Was there some clinical rationale?

-What exactly is a “National classification of disease diagnosed”?

-It appears the authors do not include all independent variables in their list. Specifically, they say malnutrition was associated with mortality in the abstract, but make no mention of that in the independent variables.

-I’m not sure of the rationale for including the difference between the various definitions of mortality in the Operational definitions section. I suggest the authors weave these into their variable definitions instead.

-It is highly unusual to include subheadings like “Competing interests” and “Acknowledgements” in the Methods. These are usually placed at the end of the manuscript.

Results:

-If the authors only included patients who had complete data, how many were excluded from their analysis? Can they say how these differed from the study population? What bias was introduced in excluding those patients? Meaning, how may they have been different than the patients included?

-How were “urban” and “rural” defined in Table 1? This should be defined in the Methods.

-Table 1: What does “Referred to higher” mean? It seems to be an incomplete phrase. Also, how were the “conditions at discharge” defined? Should be included either as a footnote to the Table or in the Methods.

-For Table 2, did the authors account for multiple diagnoses for each patient? Are these primary diagnoses?

-Table 2: What is HAAD?

-I think Table 3 could be summarized in the text. I’m not sure what the value is in seeing the differences in the patient’s sex as this does not seem to be a focus of the analysis at any point.

-Figure 1 needs some sort of scale or dots to indicate what the difference size of the lines means. It might be better displayed as a bar graph with the proportion of mortality at each discharge interval shown.

-Table 4: The variable shouldn’t be called “address”. Urbanicity would be more fitting.

-How was shock defined in this study? This needs to be clearly spelled out, particularly given the potential for subjectivity with this diagnosis. Also, what type of shock? Hypovolemic is much different than cardiogenic, for example.

Discussion:

-Paragraph 1 should not restate specifics of the Results section. Instead, I suggest the authors state in broad strokes their main findings here.

-If the authors state that mortality could be higher as this study was conducted at a referral hospital, do they have any data on referral status to justify this statement? Could that have been an important covariate to include?

-The discussion about HIV is very superficial. Can the authors say anything about whether or not the patients who had HIV were on antiretroviral therapy? Was that a covariate that could have been an important factor in HIV-related mortality?

-In general, I think the authors could improve the Discussion by including more of the implications of their findings as opposed to simply comparing and contrasting to existing literature on the topic. For example, if early mortality is a problem, what specific interventions could be employed in the future to reduce such deaths? If HIV is a problem, what could be done to reduce these deaths?

-In the limitations paragraph, what do the authors mean exactly by “under-five children admitted outside the pediatric ward”. Where were those children admitted?

Reviewer #2: In this manuscript, “Under-Five Mortality and Associated Factors in Southeastern Ethiopia,” the authors were aim to estimate the overall prevalence and associated risk factors of mortality among under-five children and post-neonates in southeastern Ethiopia. However, I feel that some adjustments and clarifications are strongly needed.

First of all, the manuscript is not well structured. The tables and figures did not present the reasonable results according to the study aim. The authors should think how to present the percentage in your tables. Then, the meaning of Figure1 was not clear.

In addition, the section of method was too weak. The authors should follow the STROBE statement to describe the study design, setting, how to select the participants, all interested variables, potential bias, the sample sizes and power, and what statistical methods used.

Finally, the authors did not indicate their study limitation.

Reviewer #3: It’s a good paper with surprising numbers in respect with European situation.

The authors should better explicit the Ethiopian health system, how the children arrive in the hospital and, for example, how long is the mean waiting time before to be visited from a medical doctor.

Particularly should be clarified the prevalence of HIV in Etiopia in the general population and in the children population or prevalence of TBC pneumonia.

The authors should specify when the therapy for pneumonia is established during the first 48 hours and specify if it was oxygen therapy or IV antibiotics.

6. PLOS authors have the option to publish the peer review history of their article (what does this mean?). If published, this will include your full peer review and any attached files.

Reviewer #1: No

Reviewer #2: No

Reviewer #3: No

---

## [Author Response · Author response to Decision Letter 0]

31 May 2021

Dear Dr Tai-Heng Chen, Academic editor of PLOS ONE, we thank you and the reviewers for a thorough reading and constructive comments on our manuscript and for the opportunity to revise and resubmit. 

We are pleased to submit the improved research article and the anonymized data. We have incorporated almost all of the comments of the reviewers and editor. 

On behalf of my co-authors, I thank you for your consideration of this resubmission. We appreciate your time and look forward to your response.

---

## [Decision Letter · Decision Letter 1]

30 Jun 2021

PONE-D-21-06094R1

Under-Five Mortality and Associated Factors in Southeastern Ethiopia

PLOS ONE

Dear Dr. Yang,

Thank you for submitting your manuscript to PLOS ONE. After careful consideration, we feel that it has merit but does not fully meet PLOS ONE’s publication criteria as it currently stands. Therefore, we invite you to submit a revised version of the manuscript that addresses the points raised during the review process.

We look forward to receiving your revised manuscript.

Kind regards,

Tai-Heng Chen, M.D.

Academic Editor

PLOS ONE

Journal Requirements:

Reviewers' comments:

Reviewer's Responses to Questions

**Comments to the Author**

1. If the authors have adequately addressed your comments raised in a previous round of review and you feel that this manuscript is now acceptable for publication, you may indicate that here to bypass the “Comments to the Author” section, enter your conflict of interest statement in the “Confidential to Editor” section, and submit your "Accept" recommendation.

Reviewer #1: All comments have been addressed

Reviewer #3: All comments have been addressed

2. Is the manuscript technically sound, and do the data support the conclusions?

Reviewer #1: Yes

Reviewer #3: Yes

3. Has the statistical analysis been performed appropriately and rigorously? 

Reviewer #1: Yes

Reviewer #3: Yes

4. Have the authors made all data underlying the findings in their manuscript fully available?

Reviewer #1: Yes

Reviewer #3: Yes

5. Is the manuscript presented in an intelligible fashion and written in standard English?

Reviewer #1: Yes

Reviewer #3: Yes

6. Review Comments to the Author

Reviewer #1: The authors have been very responsive to my comments as well as to those from the other reviewers. I think the article is significantly improved. My only comments at this time are 1) I think the authors could provide more of a rationale beyond "the variables were in the registry" as to why they were selected as predictors of mortality. There is likely other literature the authors can refer to in the Methods to indicate why things like SAM and HIV were thought to contribute to mortality. The same applies to length of stay. 2) I think the authors should mention in the Discussion or the Limitations that they did not account for any gaps in the quality of clinical care that may have contributed to such high rates of mortality.

Reviewer #3: "from 1 st August to 30 thNovember, 2019 in ..." in abstract should be changed in: from 1st September

2014 to July 2019

7. PLOS authors have the option to publish the peer review history of their article (what does this mean?). If published, this will include your full peer review and any attached files.

Reviewer #1: No

Reviewer #3: No

---

## [Author Response · Author response to Decision Letter 1]

23 Jul 2021

Dear Dr Tai-Heng Chen, Academic editor of PLOSONE, we thank you and the reviewers for the second round constructive comments on our manuscript entitled ‘’Under-five mortality and associated factors in southeastern Ethiopia’’. The comments are all valuable and very helpful for revising and improving our paper. We have tried our best to revise our manuscript according to the comments (file is attached). We are pleased to submit the improved research article. We appreciate your time and look forward to your response.

---

## [Decision Letter · Decision Letter 2]

23 Aug 2021

Under-Five Mortality and Associated Factors in Southeastern Ethiopia

PONE-D-21-06094R2

Dear Dr. Yang,

We’re pleased to inform you that your manuscript has been judged scientifically suitable for publication and will be formally accepted for publication once it meets all outstanding technical requirements.

Kind regards,

Tai-Heng Chen, M.D.

Academic Editor

PLOS ONE

Reviewers' comments:

Reviewer's Responses to Questions

**Comments to the Author**

1. If the authors have adequately addressed your comments raised in a previous round of review and you feel that this manuscript is now acceptable for publication, you may indicate that here to bypass the “Comments to the Author” section, enter your conflict of interest statement in the “Confidential to Editor” section, and submit your "Accept" recommendation.

Reviewer #1: All comments have been addressed

Reviewer #3: All comments have been addressed

2. Is the manuscript technically sound, and do the data support the conclusions?

Reviewer #1: Yes

Reviewer #3: Yes

3. Has the statistical analysis been performed appropriately and rigorously? 

Reviewer #1: Yes

Reviewer #3: Yes

4. Have the authors made all data underlying the findings in their manuscript fully available?

Reviewer #1: Yes

Reviewer #3: Yes

5. Is the manuscript presented in an intelligible fashion and written in standard English?

Reviewer #1: Yes

Reviewer #3: Yes

6. Review Comments to the Author

Reviewer #1: The authors have addressed all my remaining comments. I appreciate their responsiveness to my comments and suggestions.

Reviewer #3: (No Response)

7. PLOS authors have the option to publish the peer review history of their article (what does this mean?). If published, this will include your full peer review and any attached files.

Reviewer #1: No

Reviewer #3: No

---

## [Editor Report · Acceptance letter]

27 Aug 2021

PONE-D-21-06094R2 

Under-five mortality and associated factors in southeastern Ethiopia 

Dear Dr. Yang:

I'm pleased to inform you that your manuscript has been deemed suitable for publication in PLOS ONE. Congratulations! Your manuscript is now with our production department. 

Kind regards, 

on behalf of

Dr. Tai-Heng Chen 

Academic Editor

PLOS ONE